# Hindering and enabling factors for young employees with common mental disorder to remain at or return to work affected by the Covid-19 pandemic – a qualitative interview study with young employees and managers

**Martina Wallberg**[ID]**, Helena Tinnerholm Ljungberg, Elisabeth Björk Brämberg, Lotta Nybergh, Irene Jensen, Caroline Olsson**[ID]*

Unit of Intervention and Implementation Research for Worker Health, Institute of Environmental Medicine, Karolinska Institutet, Stockholm, Sweden

* caroline.olsson@ki.se

## Abstract

### Background

During the COVID-19 pandemic, changes in working life occurred, even in Sweden, where there was no general lockdown. The aim of this study was to examine how the COVID-19 pandemic was perceived as affecting the hindering and enabling factors among young employees with CMD to remain at or return to work, here as investigated from the perspective of young employees and managers.

### Material and methods

A qualitative design was applied with semistructured interviews with 23 managers and 25 young employees (20–29 years old). The interviews were recorded and transcribed verbatim, and the parts of the interviews related to the aim of this article were analysed using conventional content analysis.

### Results

The hindering factors were changed working conditions, decreased well-being when spending more time at home, and uncertainty. The enabling factors were decreased demands, increased balance, and well-functioning work processes. For managers it is important to be aware of warning signals indicating blurred boundaries between work and private life, to create and maintain well-functioning communication, and leave room for recovery.

**Data Availability Statement:** The datasets generated and analysed during the current study are not publicly available due to the Swedish ethical review regulation. Data are available upon reasonable request. Inquiries for data access should be sent to Karolinska Institutet, Institute of Environmental Medicine, Unit of Intervention and Implementation Research for Worker Health, Box 210, 171 77 Stockholm or Research and Data Office at the Karolinska Institute rdo@ki.se.who The Swedish Ethical Review Authority will be consulted for permission to openly share the data.

**Funding:** The Swedish Research Council for Health, Working Life, and Welfare financed this project (LN, Forte; grant number 2019-00883, https://forte.se/en/). The funder had no role in study design, data collection and analysis, decision to publish, or preparation of the manuscript.

**Competing interests:** The authors have declared that no competing interests exist.

## Conclusion

The hindering and enabling factors can be described as two sides of the same coin. Changes in the working conditions during the pandemic led to difficulties for both young employees and managers when the margins of maneuver were insufficient.

## 1. Introduction

On January 30, 2020, the World Health Organization (WHO) declared that COVID-19 was a public health emergency [1] and a threat to human health [2]. On March 11, 2020, it was declared a pandemic [3], and since then, it has resulted in fundamental changes for millions of people around the world [2].

Although many countries instigated national lockdowns to limit the spread of the virus, Sweden had a more restrained approach throughout the pandemic [2]. There was no general societal lockdown in Sweden [2], and although physical distancing was strongly recommended, it was only mandatory and enforced in certain places, such as public and private events and locales including restaurants, bars and gym facilities [4–6]. Furthermore, unlike in many other countries, Swedish kindergartens and schools for children aged 6–16 remained open throughout the pandemic [7].

In countries that instigated national lockdowns during the pandemic and where schools and kindergartens were closed, studies examining the effects of COVID-19 on working life have reported both positive and negative effects on the working conditions for employees working from home and at work.

In Sweden, employees who could work from home were strongly encouraged to do so. This measure partially aimed to limit the use of public transportation and minimise travel to protect those employees whose jobs had to be performed at their regular workplace [2], leading to approximately 30–40% of the Swedish workforce working at home [8]. A report from the Swedish Agency for Work Environment Expertise mapping the effects of the pandemic on those who were working at home reported somewhat contradictory effects on the working environment and well-being of the workers. While working from home can improve the working situation by providing increased flexibility in the balance between work and private life, which is consistent with international findings [9], it can also be a source of strain by placing the responsibility to uphold and regulate the boundaries between work and private life on the individual [8]. Accordingly, studies outside of Sweden have reported that working from home can have a negative effect on employee well-being because of blurring the boundaries between work, family and private life [10], with increased levels of stress, especially for women working from home with children [11, 12]. Another negative effect of the pandemic that is prevalent both in Sweden and other countries is increased isolation as an effect of the changed relationship between employees, here with both co-workers and managers when transferring to a digital context without the possibility of physical meetings at the workplace [8, 9]. In addition, there is the potential prolongation of working hours when working at home [12], partially because of more time spent in meetings [13], which can be associated with sickness presenteeism because of work intensification [14].

However, it was not only those who were encouraged to work from home who experienced a change in their working lives. For those working at their regular workplace (office, restaurant, hospital etc) during the pandemic entailed not only an increased risk of infecting others or becoming infected themselves, but for many it also entailed an increased workload [15]. A

so-called health debt because of an increase in work demands and work tasks and decrease in time for recovery emerged during the pandemic for those working at the workplace, specifically in human service professions in Sweden [15].

Given the changes in working life during the COVID-19 pandemic, young employees with common mental disorder (CMD), that is, depression, anxiety, adjustment disorders and stress-related ill health [16], are of special interest. A recent study showed that sick leave due to CMD was higher among young adults (20–34 years old) in Finland compared with other age groups [17], and CMD was the most common reason for sickness absence among young employees (19–29 years old) in Sweden [18].

In addition to the higher rates of young employees with CMD compared with other age groups, young people appear to have been particularly vulnerable during the pandemic. A longitudinal study in Finland on the impact of age, gender, education, living alone and telework on occupational well-being found that well-being deteriorated more for the young participants compared with the other age groups [19]. As a possible explanation, the authors of the study suggest that young people who are new to the workplace depend more on tactic knowledge and building a social network, which was negatively affected by the pandemic. In addition, it was suggested that young people were potentially more likely to work in occupations where the working conditions were affected by the pandemic to a greater extent, such as human contact professions, including healthcare and service professions. In Sweden, the employer has a major responsibility to prevent ill health and to minimise sick leave. However, systematic work environment management was hampered during the pandemic. It is therefore important to investigate managers' experiences of supervising young employees with experience of sick leave due to CMDs [20].

The present manuscript is part of a qualitative research project that initially aimed to investigate perceived causes of sick leave owing to CMD, as well as the barriers to and resources for returning to work for young employees, and to analyse the differences and similarities in the patterns of experiences of women and men from the perspectives of both employees and managers [21]. The project was designed before the outbreak of the COVID-19 pandemic, but since data collection started after the outbreak, the project could be adapted to COVID-19 and a research question was added to investigate how the COVID-19 pandemic had affected the participants' experiences of sick leave and return to work. This research question can provide an in-depth understanding of young employees' and managers' perspectives through a qualitative approach, providing a comprehensive understanding of how the pandemic affected young employees with CMD and what lessons can be learned.

The aim of the current study was to examine how the COVID-19 pandemic was perceived to affect the hindering and enabling factors among young employees with CMD to remain at or return to work, investigated from the perspective of young employees and managers.

## 2. Materials and methods

### 2.1 Design

This study is part of an ongoing, qualitative research project with the aim to apply a gender perspective on the perceived causes of sick leave owing to CMD, as well as the barriers to and resources for returning to work for young employees from the perspectives of young employees and managers [21]. Sick leave and RTW are understood as processes, where experiences that have occurred prior to, during and, where applicable, after the period of sick leave are investigated [22]. Furthermore, a gender perspective is applied to better understand similarities and differences among men and women. The current study focuses on the third research question:

- What factors related to work, home and lifestyle are identified by employees and managers as causes of sick leave?

- What resources and barriers related to work, home and lifestyle are perceived by employees and managers as enabling RTW?

- How does the coronavirus pandemic affect their experiences?

The choice of a qualitative approach in the overall project was motivated by its usefulness when the aim is to examine the experiences and perceptions of the participants and provide rich descriptions of a topic of which there is little prior knowledge [23]. The present study applied an inductive conventional content analysis in which the formulated research question was used as lens through which the participants experiences were interpreted [24]. The reason for this was that the possible impact of the pandemic on the experiences of young employees and managers of young employees with CMD to the best of our knowledge had not previously been investigated due to the novelty of the situation. The full scope of the overall project is described in a published study protocol [21].

The research project was initiated before the COVID-19 pandemic and two interview guides (one for the young employees and one for the managers) were created in accordance with the initial aim of the project in line with Patton's description of semistructured interviews [25]. The original interview guide included themes concerning experienced causes for sick leave in relation to work and in private life, previous history of sick leave, and experienced barriers and resources for RTW, as well as if and how the respondents consider age or gender to be affecting the stated experiences [21]. However, since the data collection had not begun before the outbreak of the pandemic, it was possible to add questions to the interview guide directly relating to the experiences of the pandemic and how it was perceived to affect the hindering and enabling factors among young employees with CMD to remain at or return to work. The young employees were asked about how they perceived the COVID-19 pandemic has affected their sick leave, their return to work, or ability to remain at work (at the workplace or when working from home). Accordingly, the managers were asked about their experience of working as managers for young people with CMD during the pandemic. The extensive nature of the interviews, and the conscious use of probing questions, limited the risk that the interviewees did not share their perceptions of the impact of the pandemic. Two pilot interviews were held to test the interview guide.

To strengthen the quality of the manuscript, the procedure and reporting follows both the recommendation of the consolidated criteria for reporting qualitative research (COREQ) checklist by Tong et al [26], and the standards for reporting qualitative research (SPQR) by O'Brien et al [27].

## 2.2 Data collection

Data were collected between November 2020 and August 2021 through semistructured interviews.

All interviews were conducted by CO or HTL, who both have training and experience in qualitative interviewing. Each interview lasted around one hour. The interviews were held digitally using video platforms (MS Teams and Zoom), and a few interviews were held over the phone. The interviews were transcribed verbatim and cross-checked for accuracy by HTL and CO.

## 2.3 Participants

The young employees and managers were recruited nation-wide in Sweden through a purposive sampling strategy [28], with the aim to include participants with experiences from a wide

range of occupations as well as both men and women. In total, 25 young employees were interviewed between November 2020 and May 2021, and 23 managers were interviewed between November 2020 and August 2021. There was no connection between the recruited young employees and managers.

**2.3.1 The young employees.** The young employees were recruited between October 2020 and May 2021, through ads in magazines and on Facebook, on university homepages, posters and flyers distributed by contacts of the researchers, and by OHS Centres. The recruitment material described the research project, the inclusion and exclusion criteria's, how it was possible to participate and how to contact the research team. 105 young employees contacted the research group via email and expressed their interest to participate. After that, CO and HTL called the young person to verify the inclusion criteria. For those young people who fitted the criteria, an appointment was made for an interview. Of 105 persons who had expressed an interest to participate, 74 was not included since they: did not meet the inclusion criteria (34); could not be reached (29) or came in contact when enough participants had already been included in accordance with the sampling strategy (7) or reached out after the inclusion period (4). 31 young employees agreed to be interviewed but of these (2) participants did not in the end meet the inclusion criteria or could no longer be reached (4).

The inclusion criteria for the young employees were as follows: employees aged 19–29 who currently were, or within the past two years had been, on sick leave for 3–12 weeks due to CMD, that is, depression, anxiety, adjustment disorders or stress-related ill health. The short period of sickness absence was chosen to create an opportunity to examine a group of young employees with CMD, a group that has gained less attention in comparison to long term sickness absence, and open for the possibility to reflect upon early prevention or return to work before the cases become too severe. The time frame of the recruitment period in combination with the inclusion criterion that the young employees currently were, or within the past two

**Table 1. Young employee's characteristics.**

| Age (mean, range) | (27, 20–29) |
|---|---|
| **Gender** | |
| Women | 13 |
| Men | 12 |
| **Work sector** | |
| Health care/Social services/Education | 9 |
| Service | 5 |
| White Collar | 6 |
| Blue Collar | 5 |
| **Living conditions** | |
| Alone | 15 |
| Living with parents | 1 |
| Living with partner | 7 |
| Living with a partner and child | 2 |
| **Sick leave status during the pandemic\*** | |
| Working at work after returning to work | 14 |
| Working from home after returning to work | 5 |
| On sick leave | 3 |
| Studying/Other | 6 |

\*Refers to the young employees' experiences at the time of the interviews. Some of the young employees had experiences from different situations during the pandemic. These employees are represented in multiple rows.

years had been, on sick leave meant that the participants had different experiences of either returning to work or remaining at work during the pandemic. Similarly, the managers had experiences of either having young employees either being on, returning from, or remaining at work after being on sick leave during the pandemic.

The exclusion criteria for the young employees were comorbidities that severely could affect the experiences of sick leave and RTW. Therefore, individuals suffering from posttraumatic stress syndrome, neuropsychiatric diagnoses, psychotic symptoms and bullying problems were excluded because these conditions could affect their perceptions of the causes of sick leave. Individuals without employment or employed by staffing agencies were also excluded [21].

Prior to the interviews, the young employees answered a web-based questionnaire with background questions. The characteristics of the young employees are presented in Table 1.

**2.3.2 The managers.** The managers were recruited between November 2020 and August 2021 by ads in social media, LinkedIn, on the university homepage, ads in magazines targeting managers, and by distributing posters and flyers to contacts of the researchers. 29 managers contacted the research group via email and expressed interest to participate. Five were not included since they did not work as a manager (2), or could not be reached (via mail or telephone) to book an interview (3). 24 managers agreed to be interviewed but one individual did not show up for the scheduled interview and was hence not included (1).

Managers who were eligible for inclusion were first-line managers who, during the past years at the time of recruitment for the study, had responsibility for young employees (19–29 years) who had been on sick leave due to CMD for 3–12 weeks for depression, anxiety, adjustment disorders or stress-related ill health [21]. In Sweden, employees are not obligated to account for the reason for their sick leave, but they can choose to share this information with their employer. Furthermore, the managers had to have at least one year of experience as managers and work at least half time [21]. The personal characteristics of the managers are presented in Table 2.

## 2.4 Data analysis

In the data analysis, a conventional content analysis approach was used, as described by Hsieh & Shannon [24]. A conventional content analysis is a useful approach when existing theory or

**Table 2. Manager's characteristics.**

| Age (mean, range) | (48, 36–61) |
|---|---|
| **Gender** | |
| Women | 16 |
| Men | 7 |
| **Work sector*** | |
| Health care/Social services/Education | 12 |
| Service | 3 |
| White Collar | 6 |
| Blue Collar | 3 |
| **Years of experience** | |
| –5 years | 4 |
| 6–10 years | 7 |
| 11+ years | 12 |

*Some of the managers had experiences from different work sectors during the pandemic. These participants are represented in multiple rows.

research literature of the phenomenon at hand is limited. In conventional content analysis, the coding is emergent and the categories are derived from the data [24] which means that the method is inductive and no predetermined coding scheme or theoretical framework is used.

The analysis process started with several readings of the transcripts by the research team. Then CO and HTL coded the material using NVivo software. The young employees and the managers were coded separately. In the initial coding of the collected data CO and HTL coded all content in codes capturing experiences of perceived cases of sick leave in relation to work and private life as well as hindering and enabling factors of RTW. At this stage all utterances discussing different aspects of the pandemic were coded as *Covid*. For the current manuscript, MW read the full interview transcriptions to form an understanding of the context in its whole. The material coded as *Covid* was then compared to the full transcriptions, and consensus was found in the interpretation. Thereafter, MW reanalysed the Covid data through conventional content analysis and created new, more detailed, codes of the data previously coded as Covid. Based on the Covid-data, hindering and enabling factors were identified that had been affected by Covid. The codes were then sorted into subcategories based on their relatedness, and these emergent subcategories were then abstracted into categories [24]. The preliminary subcategories and categories were discussed within the research group, drawing on the interdisciplinary knowledge of the members. The data from the interviews with the young employees and managers were analysed separately but presented jointly in the results section to provide a comprehensive picture of how the perceptions of the young participants and managers aligned and differed. Gender differences were not reported in the current study.

## 2.5 Ethics

The present study was approved by the Swedish Ethical Review Authority (registration number 2020–03271). Written and oral information about the project was given to those eligible for participation; they were informed that their participation was voluntary and that it was possible to withdraw from the study at any time without stating any reason. Informed written consent was collected from all participants.

## 3. Results

In the following section, the results from the conventional content analyses are presented. While the ways in which the young employees and the managers spoke of the hindering and enabling factors often revolved around the same topics, which can be seen in the main categories in Figs 1 and 2, they did so in somewhat different terms, which can be seen in the sub-categories. These similarities and differences will be accounted for in the following section and further reflected upon in the discussion.

The results comprise six main categories, three relating to hindering factors and three to enabling factors. The hindering factors were changed working conditions, decreased well-being when spending more time at home, and uncertainty. The enabling factors were decreased demands, increased balance, and well-functioning work processes. For an overview of the results, see Figs 1 and 2.

### 3.1 Hindering factors

**3.1.1 Changed working conditions.**   *3.1.1.1 Increased demands at work.* When reflecting upon how their ability to remain at or return to work had changed during the pandemic, some of the young employees with CMD talked about the increased demands at work. These experiences were most common for the young employees who had jobs that could not be performed from home during the pandemic. They described how their mental wellbeing had been

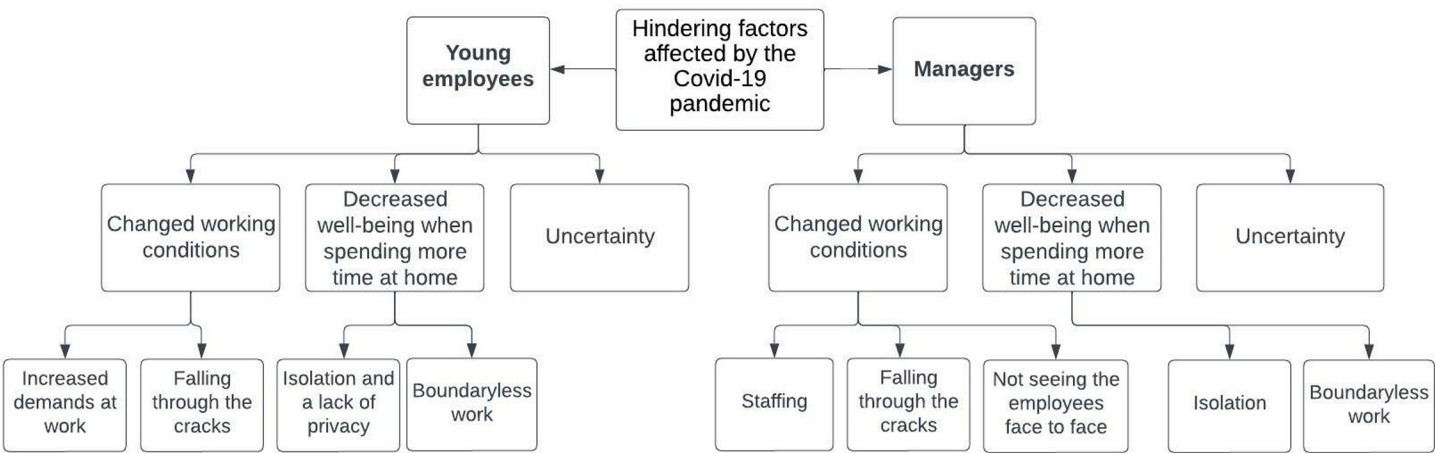

**Fig 1. Tree diagram of the hierarchal structures of the hindering factors for young employees with CMD to remain at or return to work affected by the COVID-19 pandemic.**

negatively affected when they had to work long shifts in what they perceived to be uncertain conditions. On a similar note, the strain of having to cover for their colleagues who were ill, without receiving substitutes was perceived to limit the possibility for recovery at work for those had returned to work after being on sick leave due to CMD. This increased intensity at work, exaggerated by the uncertainty following the spread of the pandemic, was perceived to be extra stressful for the young employees with CMD trying to remain at work:

> *And it became more now also considering the pandemic, so it was . . . because there are many who are sick, and we cover for people who are sick as well. Therefore, then the pressure only increases. / . . ./. And it has been like that in previous periods. Now, it has been a very long period when there has been a lot of sick leave every week. So I think it may have something to do with it, too, that was like the last straw in some way, or it was too much because there was no recovery. I never felt that . . . normally, then I was very tired . . . or before, then I was very tired during the weekend; but then, I could sleep and wake up on Saturday and just feel, 'Oh, God. At least I got a good night's sleep', and so I felt rested. In recent months, then, it has been*

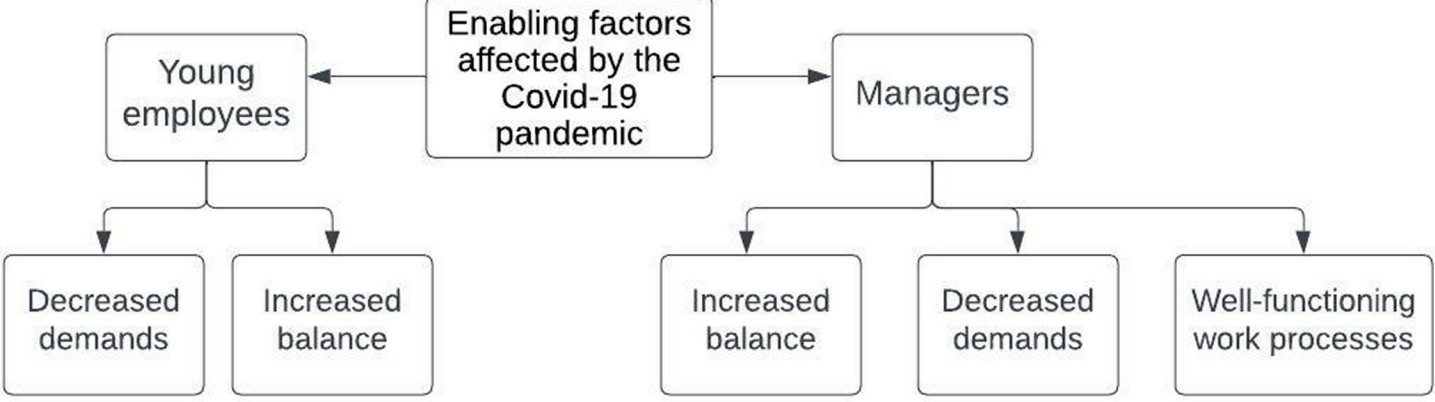

**Fig 2. Tree diagram of the hierarchal structures of the enabling factors for young employees with CMD to remain at or return to work affected by the COVID-19 pandemic.**

*that I never felt rested. I was constantly tired. Yes.* (IP 22, young man, building caretaker, blue-collar worker)

The changed working conditions and increased intensity had also created new sources of conflicts at the workplace because of the uncertainty regarding new routines and working methods. One of the young employees described how this added to the strain of remaining at work after being on sick leave due to CMD.

*3.1.1.2 Staffing.* While the increased demands at work for young employees with CMD caused by the pandemic were also addressed by some of the managers., they mainly discussed the matter in terms of staffing issues. When the sick rates among their inhouse employees increased due to the pandemic, difficulties arose in getting substitutes with the right skills and finding enough personnel to supervise the substitutes since they could not supervise new staff themselves. One of the managers recognized that the issues relating to staffing increased the work intensity for their regular employees. This in turn was perceived limited the possibility to ease the workload for the young people with CMD who had returned to work after being on sick leave, or that were at risk for not being able to remain at work. While the young employees might have needed the option to work at a slower pace, one manager concluded that the additional tasks that arose due to the pandemic tended to fall on these young employees with CMD who already took on a great deal of responsibility:

> *And I, as a manager, who is so dependent on everyone who has the strength to deliver all the time, unfortunately loses that, 'Shit, I should not burden her, I should not believe that she is eager and wants to do a lot' because I know it will backfire. So I'm probably thinking that it might result in a period of sick leave in the autumn, which it might have happened anyway.* (IP 16, female manager, social service)

Furthermore, the managers perceived the work to be affected in a negative way when the regular staff needed to train the substitutes, and one of the managers said that they had to cut down on internal training and education for the employees because of a staff shortage.

*3.1.1.3 Falling through the cracks.* Another effect of the pandemic that the young employees with CMD perceived as hindering for their possibility to remain at or return to work was the increased risk of falling through the cracks due to the pandemic. One of the young employees talked about how it had become more difficult to get in touch with the closest manager during the stressful work situation caused by the pandemic. On a similar note, another young employee described how the follow-up meetings during the return-to-work process with the closest manager had been cancelled due to the intensified work situation. The young employee continued to describe how the pandemic was perceived to have created more critical managerial tasks than the scheduled meetings, and that the situation refrained him from asking the manager for more regular check-ins. This increased risk of falling through the cracks and the perceived obstacles for asking for help led to the young employees with CMD not receiving the help they needed to remain at work. One young employee concluded that:

> *Or like the fact that the manager was basically never present. That . . . now on the other hand she was . . . or now she is the head of both . . . for two hospitals, if you say . . . but yes, you would have seen her once a week top. So that made it very difficult to get hold of her that, yes, to talk if there were things you needed to bring up or such.* (IP 6, young man, occupational therapist, healthcare)

*3.1.1.4 Not seeing employees face to face.* Similarly to the young employees experiences of the increased risk of falling through the cracks, the managers recognised that there had been a diffusion of responsibility during the pandemic regarding the handling of the young employees with to CMD. But while the managers also talked about these matters, they mainly did so in terms of the difficulties that arose when they could not meet their young employees in person and see how they were doing. The managers described that it was more difficult to get an understanding of how their young employees with CMD were coping when they did no longer meet them in person due to the restrictions during the pandemic. Before the pandemic, cues regarding their young employees' mental well-being were picked up during small talk or when eating together in the lunchroom. But for many managers this was not possible during the pandemic when their young employee had to work from home, and one manager described how it was more difficult to truly understand how her young employees with CMD were coping:

> *But there have been some meetings and stuff where, I think it gets wrong sometimes when you do not see the person. Because often, things happen to the body, posture and gaze before you say something. /. . ./ I could never get this down on paper. But it's just this feeling you get when you have a dynamic with a person in a room. And many who have lived with mental illness, perhaps since they were little, are almost just like addicts, extremely good at hiding things until it just does not work anymore. So, it is not like . . . Most people have so much poker face until you find the right button and you get through. And I think you miss out on that in the pandemic.* (IP 20, female manager, group manager rehabilitation)

Accordingly, some of the managers expressed feelings of frustration and stress over the difficulties of tuning in with, and being perceived as available for, their young employees with CMD when work was carried out remotely due to the pandemic.

The young employees' experiences of the increased difficulty of getting a hold of their manager were also reflected in the interviews with the managers. Some of the managers talked about how it was difficult to ensure that everyone received the same information when the working groups were no longer located in the same place when working from home or not being permitted to gather in full group due to the pandemic. This was in turn perceived to affect the young employees with CMD to a greater extent. Some of the managers described how they had become less involved in their young employees' working lives when they were physically separated due to the pandemic, which made it more difficult to keep up to date with the young employees' ability to remain at or return to work. Many of the managers believed that e-mails, digital meetings, or telephone calls could not completely replace physical meetings when it came to working with young employees with CMD.

**3.1.2 Decreased well-being when spending more time at home.** *3.1.2.1 Isolation and a lack of privacy.* Among the young employees with CMD, feelings of isolation caused by the COVID-19 pandemic were recurringly described as a strain that was perceived to affect their overall well-being and ability to remain at work. The feelings of isolation were partially because of the societal recommendation in Sweden during the pandemic that those who could work from home were strongly encouraged to do so. Some of the young employees working from home talked about the limited possibilities of meeting and interacting with their co-workers and how the increased isolation deteriorated their mental well-being:

> *It's very exhausting, I think, to work a lot from home, not have a like . . . a context. I've just started a new job, and I do not know my new colleagues, and we meet only digitally; it's really weird! /. . . / Ehm, it's like exhausting that. . . to live as much in your own head as you do*

*when you . . . walk around by yourself at home, all day . . .* (IP 3, young female, communications officer, white-collar worker)

However, it was not only working from home that resulted in increased feelings of isolation. Some of the young employees with CMD also described how societal restrictions limited the possibility to visit families and relatives who lived in other cities. At the same time other meeting places, such as gym facilities and community meeting places, were shutting down. These different limitations of some of the young employees social interactions increased their feelings of isolation, which in turn was perceived by some to increase their social anxiety. One of the young employees described it in the following way:

*/. . ./an important part of processing social anxiety is to train and be out among people and so on. But it has not been possible, so the social anxiety has become very, very strong during the corona.* (IP 26, young man, building caretaker, blue-collar worker)

The managers also recognised that the pandemic had increased the isolation for their young employees with CMD. Many of the managers discussed different aspects of being young that they considered to be affected by the pandemic, and that they furthermore perceived to affect the young employees with CMD to an even greater extent. Among these were the increased risk of isolation for their young employees with CMD who had their families living in another city. Being young in a new city and working from home were generally perceived by the managers as increasing the risk of isolation. The managers described the increased risk of negative effects for those with CMD in a similar way to that of the young employees. In accordance with the young employee who felt that his social anxiety had increased when working from home, one manager discussed how the lack of social contact when working from home could lead to a decrease in the sense of belonging at the workplace. Another manager reasoned that the young employees with CMD who worked from home due to the pandemic lost the opportunity to tune in with other colleagues in the hallways, which could harm their well-being and affect their ability to remain at work. Here, one of the managers reasoned that it is difficult for young employees to understand what is and can be expected from them when they work from home, and that this uncertainty can be extra straining for those who already struggle with CMD.

*And when you sit at home, to know 'what is expected of me?' and 'what is good enough?' and that sort of thing. Because you have no colleagues to discuss with. So, it is . . . And you do not become a part of the work gang, the collegial team, in the same way.* (IP 23, female manager, state and municipal administration)

However, feelings of isolation and an increased uncertainty of what is expected were not the only downsides to spending an increased amount of time at home. The young employees who lived with another adult and/or a child also talked about feelings of being trapped and not being able to have time alone. The lack of privacy affected the ability to rest for those young employees with CMD who were still recovering from their sick leave. In addition to limiting the possibility for privacy, some of the young employees working from home in small spaces talked about the difficulties in disconnecting from work that arose due to the inability to switch environments when the workday was over.

*3.1.2.2 Boundaryless work.* Another effect of the pandemic that some of the young employees with CMD who were working from home experienced was the sudden lack of clear boundaries between work and private life. For some of the young employees, difficulties with

maintaining the distinction between work and private life had been a contributing factor to their sick leave or difficulties to remain at work before the pandemic as well. This issue prevailed during the pandemic, and some felt that working from home blurred the boundaries even more and led them to work around the clock without breaks. One young employee described that initially, the situation was perceived to be beneficial because suddenly, time that previously had been spent on travelling to or from work, having coffee with colleagues, or taking breaks could be solely spent on work:

> /. . . / And then, we started working from home. And back then, I thought it was really nice because you were not disturbed by a lot of questions, and you saved a lot of time because you had no travel time to and from work and the office and so on. And you could have lunch in ten minutes and then continue working. And you did not have to take these breaks and you could basically work from eight to twelve nonstop. So I worked very hard, and you didn't get . . . It was a very uncertain situation with corona and so on. So it was almost like a lockdown. That there was nothing to do. So I worked all weekend, and I worked around the clock, you could say. (IP 19, young woman, accountant, white-collar worker)

The young employee continued to describe how the situation eventually became unsustainable. She concluded that even though those coffee breaks with colleagues seemed unnecessary at the time, they proved to be important for maintaining her well-being at work.

One of the managers also addressed the issue of employees working around the clock as a problematic effect of working from home due to the pandemic, and discussed the potential impact on the young employees' with CMD's mental well-being. As a measure of dealing with the increased risk of blurred boundaries, the manager emphasised the importance of having regular check-ups and reminding the young employees to take regular breaks when working from home. The manager further emphasized the increased importance of being aware of subtle warning signals of blurred boundaries between work and private life, such as acknowledging at what time of the day an email was sent.

**3.1.3 Uncertainty.**   Both the managers and young employees discussed how the circumstances during the pandemic had awoken feelings of uncertainty. This uncertainty was in turn perceived to affect the young employees ability to remain at work

The young employees with CMD described how the overall uncertainty stemming from the pandemic affected their mental well-being. Not knowing what would happen during the pandemic, or what would follow, was perceived to increase the levels of stress, anxiety, and catastrophic thoughts that the young employees with CMD already had, and that negatively affected the young employee's ability to remain at work. Similarly, some of the young employees struggled with feelings of having a lack of control over the situation during the pandemic, and one young employee described this as follows:

> So this not being able to plan in the same way in the job I have, that a lot of things are very uncertain, and for me, it is . . . it also triggers my anxiety as well, that this . . . not having control over the situation, so then I feel worse in periods like, so. So that is frustrating [laughter] right now. (IP 2, young woman, employment consultant, social services)

The young employees' mental-well-being was furthermore affected by the increased financial stress and a fear of losing one's job that was caused by the changes on the Swedish labour market during the pandemic. These stressful thoughts were perceived to have negatively affected their ability to remain at work during the pandemic. The feelings were further

perceived to be exaggerated by the increased stress caused by the continuous news updates and difficulties in taking in and following the changing societal restrictions during the pandemic.

The managers also discussed how the constant need to provide new information regarding the situation caused stress not only for themselves, but for their young employees as well. While the managers recognized that the uncertainty affected all employees, it was perceived to be an extra burden for the young employees with CMD. The recognition of the extra burden that the uncertainty put on the young employees with CMD was also recognized in the increased fear of getting sick during the pandemic:

*What the pandemic has done /. . ./ well, initially, it was a lot that we needed to provide new information all the time; we needed to check everything. Then, there was this fear, in the sense of what happens if the customer becomes ill and what happens if the whole group becomes ill.* (IP 1, female manager, healthcare)

The fear of getting sick, or bringing the disease home with them after their shift, was perceived by the managers to be extra burdensome for the young employees with CMD as it increased their anxiety that was already present before the pandemic.

### 3.2 Enabling factors

**3.2.1 Decreased demands.**   While some of the young employees with CMD talked about how the pandemic had affected their mental well-being in a negative way due to a lack of social contact, it was not always the case. Some young employees described how the reduced social contact both at work and outside of work had made it easier to remain at or return to work.

One young employee reasoned that even if the restrictions during the pandemic sometimes had her feeling a bit lonely, the limited social contact in her private life had also given her sufficient time for recovery and rest. Another young employee described how the pandemic had made it easier for her to let go of the responsibilities in her personal life. Some of the young employees talked about how it was difficult to balance the perceived expectations on social interaction from their surroundings with their increased need for rest during or after being on sick leave. When these expectations were no longer as tangible due to the pandemic, keeping a busy social schedule was not an option:

*But I think it's a lot that I haven't had to partake in social interactions. I think it's so incredibly, incredibly fun to meet people and so on, but I also notice how much it drains my energy and my strength. So that I think it has been a bit of a rescue in some ways.* (IP 16, young woman, real estate manager, white-collar worker).

Overall, many of the young employees described how the perceived demands in their private lives had been reduced during the pandemic. This resulted in an increase in time for rest and recovery that affected some of the young employees' ability to remain at or return to work in a positive way.

The decreased social demands experienced by some of the young employees were also prevalent at work. While some young employees had experienced increased demands at work, others had the opposite experience. One of the young employees who was returning to work after being on sick leave concluded the following:

*And this is something you almost dare not say, but now I'll do it anyway. For me, it was not so bad that it became . . . okay, the sick leave overlapped with the fact that we had a lot less to do. It was a very soft start, and I would not have had that soft start if things were normal.*

*Then, it would have been right into the wall, yeah, right into the tiles, keep going.* (IP 9, young woman, occupational therapist, healthcare)

Accordingly, other young employees who were also working at their regular workplace during the pandemic described how the pandemic had resulted in reduced activity at work. This made it easier for them to keep up at work without using up all their energy.

Another factor that some of the young employees perceived to ease the demands at work was the clarifications regarding which rules applied to being ill and staying at home. During the pandemic, Swedish governmental guidelines stated that anyone who experienced cold symptoms should stay at home. And although the guidelines targeted physical symptoms, it was perceived to make it easier to stay at home for other reasons as well. One young employee described how the guidelines made it feel more ok to stay at home and take a sick day due to her mental health, since taking a day off was not questioned in the same way by her colleagues when everybody had to stay at home when they were not feeling well:

*But it was a little nice, actually, uh, because all these colleagues who sort of go to work when they are sick, they kind of had to stay home at the very first /. . ./ And then it became very clear /. . ./ This is how it is, well, am I sick enough to stay at home.* (IP 1, young woman, after school leisure leader, education)

This changed approach to being sick was also recognised by the managers. Some described it as a new type of more open and understanding communication that went beyond cold symptoms and physical illness. One manager described how the pandemic had created an increased empathy among the employees for those who called in sick had, whether it was because of cold symptoms or not. The change in attitude towards staying at home during an illness was perceived to make it easier for young employees with CMD to work in accordance with their capability.

**3.2.2 Increased balance.**   Although some of the young employees and managers described how the boundaries between work and private life had been blurred when working from home due to the pandemic, others felt that it had increased the balance between work and private life.

Some of the young employees described how it was easier to work while at the same time taking time for recovery when working from home. One employee described how the time saved on commuting to work instead could be spent on other activities that maintained the mental well-being. This was also discussed by another young employee, who felt that it was easier to balance parenthood when spending a larger part of the day at home which decreased some of the stress when returning to work after being on sick leave. One of the young employees whose return to work took place remotely due to the pandemic described how it was easier to keep up with the good habits she had implemented during her sick leave when working from home. She described how it was easier to take a break and go outside to get some fresh air and clear the head during the workday, and to adjust the work schedule to one's own needs and energy level when working from home:

*And especially for me, who . . . where it is . . . like, food and mealtimes are so important . . . Eh . . . that . . . that I can . . . eat, even if I have a meeting . . . at one o'clock, when I eat my lunch, I can eat my lunch, then . . . and I would not have been able to do that if we were sitting physically! Because it's a . . . or, well. Of course, I could have, but I would have felt that it was much harder to . . . to handle that. Because now I can do it because no one else cares! I can still attend the meeting.* (IP 11, young woman, management consultant, white-collar worker)

The positive effects that the reduced time commuting to work and increased balance between work and private life had on their young employees' mental health were also recognised by one of the managers, who described how his young employees with CMD seemed to be less stressed when working from home.

**3.2.3 Well-functioning work processes.**   In the interviews with the managers, it was articulated that the uncertainty and negative effects following the pandemic appeared to be somewhat eased in those organisations with well-functioning work processes for communicating with their employees during the pandemic. This was the case for the organisations that had employees working on site, as well as organisations where most of the employees were working from home. And while these work processes typically targeted all employees, the managers also discussed how they had made it easier to stay on top of the cases of the young employees with CMD that were either on sick leave, returning to work, or that had returned to work after being on sick leave.

One manager whose employees were working mainly at their regular workplace described how she had experienced the employees in other groups within the company being more anxious than the employees in her group. When reflecting upon this difference, the manager concluded that her group had been quick to provide the necessary safety equipment in the early stage of the pandemic to make the employees feel safe. Another measure of ensuring feelings of safety described by those managers who had employees working at their regular workplace was to have a continuous dialogue with the employees to make them feel involved in the changes occurring because of the pandemic:

> *So, they have been strong anyway during the pandemic, and I think that is because we have tried to have a lot of dialogue, and I have tried to go out with weekly newsletters and clear information at . . . frequent, every week. Not too much and not too little, but what is absolutely necessary. And the staff knew that.* (IP 14, female manager, head of unit, social service)

Having a proactive, continuous, and ongoing dialogue with their employees was also perceived as an important measure to gain an understanding of the situation and mental well-being of their young employees for those managers whose employees were working mainly at home. In line with this, many of the managers whose employees were working remotely highlighted the importance of having ways of 'seeing' their young employees digitally and staying up to date on their current situation both personally and professionally. This was done in several ways, from maintaining informal communication through digital coffee breaks with co-workers to managers dedicating time to their digital calendars for spontaneous contact with employees. Furthermore, one of the managers talked about different solutions to keep up to date with employees:

> *We do a check-in, where everyone checks in on a number and tells how they feel, both professionally and privately, to keep track of each other. And if someone checks in on a four or five, then I always call that person and hear a little more. So it is a very good tool.* (IP 10, female manager, private company, service sector)

In addition to communicating with their young employees with CMD to stay updated on their situation, some of the managers also recognised the importance of having a continuous dialogue with other instances internally, such as the human resources office (HR), to catch cases of early symptoms of mental health issues.

## 4. Discussion

This study sought to examine how the COVID-19 pandemic was perceived to affect hindering and enabling factors among young employees with CMD to remain at or return to work, investigated from the perspectives of young employees and managers. In the following section, the results are synthesised and discussed.

The young employees and managers reflections regarding the hindering factors for young employees with CMD to remain at or return to work affected by the pandemic revealed both similar and different perceptions. Both the young employees and managers talked about the intensification of the work situation. An increase in demands and stress for employees working at work during the pandemic have been shown in previous research of work during the pandemic as well [15, 29]. However, the ways in which the young employees and managers in this study described this perceived intensification differed. The changed working conditions did not only pose a hinder due to an increase of the workload that limited the managers possibilities to uphold a slower work pace for their young employees with CMD. The young employees talked about how these rapid and unexpected changes had also led to increased feelings of uncertainty and anxiety, which was perceived to deteriorate their mental well-being. In this case, the negative effects of an increased workload could possibly have been eased if they were not also connected to an uncertainty of how the pandemic would affect their work and lives that in turn worsened the young employees' anxiety. Similarly, while the managers recognized that the limitations in seeing their young employees with CMD in person and keep up to date on their cases increased the risk of them missing out on how they were coping, the young employees also talked about how one hinder caused by the limited social contact with their managers was that they didn't feel prioritized and refrained from seeking out support. The difficulty in upholding well-functioning communication paths with their employees has been a reported managerial challenge during the pandemic [30], and these results indicate that it not only causes stress for the managers, but also affects the ability of young employees with CMD to remain at work.

The negative effects found in previous research of both being young at work and working from home during the pandemic, such as feeling isolated [31–34], struggling with upholding boundaries between work and private life [10], not being able to connect and network with colleagues at work [19], and living in a small space that did not provide fundamental prerequisites to work [35], were all perceived to pose an extra burden for the young employees with CMD both due to their age and mental health. And although some of the hindering factors discussed by both the young managers and employees occurred in the young employees' lives outside of work, such as societal restrictions limiting the possibility to visit their families or partaking in social activities outside of work, it affected their ability to remain at work since it deteriorated their overall mental well-being.

When reflecting upon the enabling factors affected by the pandemic, some of the young employees and the managers talked about how working from home could increase the balance between work and private life. Similar results in previous research has shown that for some, telework can result in an increased balance in life [9]. Accordingly, while some young employees with CMD in this study struggled with working from home, other benefited from the increased opportunity to adjust their workday to their capability. This actualizes the importance of individual adaption of the return-to-work process and encourages active reflection on what that process can look like. The importance of acknowledging and adjusting to individual conditions in the return-to-work process has previously been recognized in the discussion of the concept of margin of maneuver [36]. The concept refers to the possibility for the individual to meet the production targets without it having negative effect on her health by adjusting the

work conditions [36] and illustrates how different situations can be either hindering or enabling depending on the margin of maneuver. In a similar way, the results in the present study show how the hindering and enabling factors can be described as two sides of the same coin. The rapid changes in the working conditions during the pandemic led to difficulties for both young employees and managers when the margins of maneuver were insufficient. This was not only the case for the effects of working from home for young employees, but for the demands that they experienced during the pandemic as well. While some young employees experienced an increase in demands, others talked about how the reduced social demands both at work and in the young employees' private life had created room for recovery and rest that was not there before the pandemic. Another aspect that appeared to reduce the demands during the pandemic was the increased empathy and acceptance among co-workers towards illness that was not only directly related to the pandemic. The young employees with CMD described how this change of organizational culture towards one where it was perceived to be more socially acceptable not to always perform at one's peak had made it easier for them to work at their own capacity and not push themselves beyond their limits. Previous research has made a connection between organisational culture and sick presenteeism [37], and this study further brings forth the importance of paying attention to how young employees with CMD are affected by the attitudes towards being ill, no matter the cause.

## 5. Conclusion

Even if the present study was conducted during the pandemic, with the extraordinary circumstances it entailed, there are still lessons to be learned for those working with young employees with CMD after the pandemic as well. Some changes are expected to prevail [31] while more general factors directly related to being a young person with CMD at work likely will not change. The findings of the current study show that working from home as a young employee with CMD can lead to an increased balance between work and private life which in turn can ease the process of returning to or remaining at work. However, as a manager, it is important to be aware of possible warning signals indicating blurred boundaries between work and private life. This emphasizes the importance of creating and maintaining well-functioning communication between managers and young employees with CMD. Furthermore, as a manager of young employees with CMD, it is important to both encourage and leave room for rest and recovery. The findings of this study show that it can be important to consider how the organisational culture and attitudes towards illness and sick leave affects the young employees' ability to remain at work.

## 6. Strengths and limitations

The current study followed the published study protocol without any major revisions [21]. However, one limitation is that the inclusion criteria stating that young participants should have been on sick leave within the last year had to be prolonged to include the last two years prior to the interview due to initial recruiting difficulties.

Because of the pandemic, the interviews had to be held digitally using video platforms (MS Teams and Zoom), and a few interviews were held over the phone. The interviews were initially planned as face-to-face interviews, but conducting them digitally facilitated nationwide recruitment. Thus, a key strength of the current study is the representation of different perspectives among the participants. The nationwide recruitment of managers and young employees from different industries facilitated the representation of different viewpoints. Furthermore, the participants had different experiences during the pandemic. Some young employees were working from home, while others worked at their regular workplace. The

managers were working either from home or at their regular workplace, as well as having employees working both from home and at the regular workplace.

The analysis of the collected data was strengthened by the interdisciplinary collaboration in the research group. The variation of disciplinary backgrounds, including sociology, gender studies, occupational medicine, and occupational health, provided different viewpoints and perspectives [27].

Measures have been taken to strengthen the transparency of the study and recommendations following STRQ [27] and the COREQ checklist [26] were taken into consideration when presenting the results of the current study. Thereby, we have strived to increase the transferability of the results to other contexts. Even though the pandemic constituted an extraordinary situation with contextual factors that affected the work [38, 39], including different restrictions and overall uncertainty of what would come, it also provided us with a unique opportunity to examine the future of work now. One important notion in the research on work and COVID-19 is that there is a difficulty in transferring findings from studies during the pandemic because the situation was extraordinary, with circumstances specific to the prevailing situation [35]. However, with schools and other parts of the community remaining relatively open in Sweden under the pandemic, the changes in working and private life were not as extensive as in other countries. Therefore, it is possible to reason that the situation in Sweden during the pandemic can be considered more comparable to the 'new normal' [7].

## Acknowledgments

The authors gratefully acknowledge the young employees and managers who participated in the study.

## Author Contributions

**Conceptualization:** Helena Tinnerholm Ljungberg, Elisabeth Björk Brämberg, Lotta Nybergh, Jensen Irene, Caroline Olsson.

**Formal analysis:** Martina Wallberg, Helena Tinnerholm Ljungberg, Caroline Olsson.

**Funding acquisition:** Lotta Nybergh.

**Investigation:** Caroline Olsson.

**Methodology:** Helena Tinnerholm Ljungberg, Elisabeth Björk Brämberg, Lotta Nybergh, Jensen Irene, Caroline Olsson.

**Project administration:** Martina Wallberg, Caroline Olsson.

**Writing – original draft:** Martina Wallberg.

**Writing – review & editing:** Martina Wallberg, Helena Tinnerholm Ljungberg, Elisabeth Björk Brämberg, Lotta Nybergh, Jensen Irene, Caroline Olsson.

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
