## [Decision Letter · Decision Letter 0]

1 Nov 2022

PONE-D-22-17806Hindering and enabling factors for young employees with Common Mental Disorder to remain at or return to work affected by the Covid-19 pandemic – a qualitative interview study with young employees and managersPLOS ONE

Dear Dr. Pettersson,

Thank you for submitting your manuscript to PLOS ONE. After careful consideration, we feel that it has merit but does not fully meet PLOS ONE’s publication criteria as it currently stands. Therefore, we invite you to submit a revised version of the manuscript that addresses the points raised during the review process.

We look forward to receiving your revised manuscript.

Kind regards,

Nabeel Al-Yateem, PhD

Academic Editor

PLOS ONE

Journal Requirements:

Reviewers' comments:

Reviewer's Responses to Questions

**Comments to the Author**

1. Is the manuscript technically sound, and do the data support the conclusions?

Reviewer #1: Partly

2. Has the statistical analysis been performed appropriately and rigorously? 

Reviewer #1: N/A

3. Have the authors made all data underlying the findings in their manuscript fully available?

Reviewer #1: No

4. Is the manuscript presented in an intelligible fashion and written in standard English?

Reviewer #1: Yes

5. Review Comments to the Author

Reviewer #1: The introduction logically leads to the study objective. The objective was “… (to) examine how the COVID-19 pandemic was perceived to affect the hindering and enabling factors among young employees with CMD to remain at or return to work, investigated from the perspective of young employees and managers.”

For the methodological section, please add the methodological orientations of the study. Based on this orientation please justify why a conceptual framework was not used or present the conceptual framework.

The COREQ criteria are used, however, some of the information on the methodology refers to a published article. However, it was not possible to retrieve the information, since the reference is blinded.

For the interview guide, themes addressed within the absence and return experience need to be added.

To be included, workers had to have a CMD in the last 2 years. If recruitment started in 2020, this means that some participants had a sick leave before the pandemic. Thus, how do these experiences compared to the others? Also, participants had to have between 3 and 12 weeks of absence from work. These represent non-complex cases of absence. Thus, these cases are not known to drive the most burden to the individual or the society. Is it possible to justify this criterion?

In a qualitative study, participants’ characteristics and their context are usually described more in depth. Also, based on the Finnish study they mentioned, is it possible to add gender, and education? Also, what was the sick leave status of the participants? For managers, was the years of experience collected and how many employees under the supervision of the supervisors? This could influence the experience with CMD workers.

Information on how participants were recruited should be added.

In the data analysis section, the authors mentioned that “the utterances discussing different aspects of the pandemic were coded as Covid (…) who thereafter reanalysed the Covid data through conventional content analysis”. Based on this, if I understood correctly, only the content coded COVID was taken and analysed for this study. In the data collection section, it is mentioned that the interviews were done during the pandemic. Thus, it is unclear why COVID was coded if the experience is within a covid context. The risk is to assume that covid did not have an impact, just because someone failed to talk about covid at one point in the interview. We cannot assume this. More information on this is needed to assess possible bias during coding.

Is the coding emergent or based on a framework?

The authors mentioned that an interdisciplinary team of researchers did the analyses, please clarify which disciplines?

Another point needs clarification: How do managers know that the employee they supervise was absent for a CMD. Is the reason for absenteeism confidential in Sweden?

In the results section, it is not always possible to see what caused the absence, maintained the absence or made it difficult to return or sustain work. Young's (2010 in JSTOR) framework would help structure this information.

Most of the results describe what most people had to live with during the pandemic. Thus, the mechanisms particular to young workers having CMD and supervisors having workers with CMD is not clear. Consequently, it is not clear what the manuscript adds to the existing knowledge.

The presentation of the results could be revised in order to present the results in more depth, among other things by looking more closely at the links between increased balance and decreased demands, increased demands at work and low staffing. Uncertainty appears to be in part underneath the work demands and low staffing. What these results appear to highlight is the concept of the margin of manoeuvers in work disability (see Durand et al. 2016 Handbook of return to work). It can thus be an enabling factor when a margin of manoeuvers exists or an obstacle when the margin of manoeuver is insufficient. Having a low margin of manoeuvers for managers may have contributed to the employee falling through cracks.

The citation of IP 19 (lines 351-354) covers the themes of isolation, uncertainty and to some extent the boundaries. The 3 aspects are possibly linked, but the manuscript fails to provide a more in-depth understanding of how factors interrelate. This supports the need to add more links between factors.

Concerning the distinction between work and family and the proximity of the family. Based on gender differences known in the literature, was the family proximity more problematic for men participants?

The citation lines 477-479 (“So, they have been strong anyway during the pandemic, and I think that is because we have tried to have a lot of dialogue, and I have tried to go out with weekly newsletters and clear information at … frequent, every week. Not too much and not too little, but what is absolutely necessary. And the staff knew that. (IP 14, female manager, head of unit, health care)) does appear to be a managerial strategy in general, and not in line with young workers having CMD. It is not always clear if managers talk about managing in a covid context or managing young workers having CMD in a covid context.

The discussion presents three interesting findings. However, these findings are not the main findings, as these did stand out in the results. As an example, in line 513 “….could imply difficulties for managers in understanding and recognising certain challenges for young people with CMD to remain at or return to work”, this was not clear for me after reading the results.

6. PLOS authors have the option to publish the peer review history of their article (what does this mean?). If published, this will include your full peer review and any attached files.

Reviewer #1: No

---

## [Author Response · Author response to Decision Letter 0]

16 Dec 2022

Journal Requirements: 

Important: If there are ethical or legal restrictions to sharing your data publicly, please explain these restrictions in detail. Please see our guidelines for more information on what we consider unacceptable restrictions to publicly sharing data: http://journals.plos.org/plosone/s/data-availability#loc-unacceptable-data-access-restrictions. 

Note that it is not acceptable for the authors to be the sole named individuals responsible for ensuring data access.

Authors response: The datasets generated and analysed during the current study are not publicly available due to the Swedish ethical review regulation. Data are available upon reasonable request. Inquiries for data access should be sent to Karolinska Institutet, Institute of Environmental Medicine, Unit of Intervention and Implementation Research for Worker Health, Box 210, 171 77 Stockholm or contact the principal investigator Caroline Olsson, caroline.olsson@ki.se, who will then contact the Swedish Ethical Review Authority for permission to openly share the data.

Author’s comment: We have double checked and the list of the affiliations in the manuscript is correct.

Reviewers' comments:

Reviewer's Responses to Questions

Comments to the Author

1. Is the manuscript technically sound, and do the data support the conclusions?

Reviewer #1: Partly

2. Has the statistical analysis been performed appropriately and rigorously? 

Reviewer #1: N/A

3. Have the authors made all data underlying the findings in their manuscript fully available?

Reviewer #1: No

4. Is the manuscript presented in an intelligible fashion and written in standard English?

Reviewer #1: Yes

 

5. Review Comments to the Author

Reviewer #1: The introduction logically leads to the study objective. The objective was “… (to) examine how the COVID-19 pandemic was perceived to affect the hindering and enabling factors among young employees with CMD to remain at or return to work, investigated from the perspective of young employees and managers.”

Note: When the authors response refers to specific lines in the Tracked changes manuscript, it refers to the line numbers when the changes are made visible

Reviewer comment: For the methodological section, please add the methodological orientations of the study. Based on this orientation please justify why a conceptual framework was not used or present the conceptual framework.

Authors’ response: The methodological orientation of the study has been added in (lines 144 to 148) and more clearly defined as an inductive conventional content analysis. In these lines, it is now added that due to the novelty of the situation during the pandemic, an inductive approach was deemed appropriate to explore the experiences of the participants. 

Reviewer comment: The COREQ criteria are used, however, some of the information on the methodology refers to a published article. However, it was not possible to retrieve the information, since the reference is blinded.

Authors’ response: Thank you for this comment. More information will be available when the reference is no longer blinded, but we have also added information regarding the overall project (in lines 138 to 140), the methodology of the overall project and the present study (in lines 141 to 149), and the interview guide (in lines 150 to 163) to make the manuscript more independent from the study protocol. 

Reviewer comment: For the interview guide, themes addressed within the absence and return experience need to be added.

Authors’ response: The interview guide is now more thoroughly described in lines 150 to 163 to clarify its different themes relating to both the absence and return experiences as well as the perceived effects of the pandemic.

Reviewer comment: To be included, workers had to have a CMD in the last 2 years. If recruitment started in 2020, this means that some participants had a sick leave before the pandemic. Thus, how do these experiences compared to the others? 

Authors’ response: The time frame of the recruitment period made it possible to explore both the experiences of returning to work during the pandemic (for those who were on sick leave prior to or during the pandemic), and the experiences of remaining at work during the pandemic (for those who had a sick leave prior to, or during the pandemic), thus addressing the objective of this study, which was to examine how the COVID-19 pandemic affected factors for both remaining at or returning to work. This has been clarified in lines 215-220.

Reviewer comment: Also, participants had to have between 3 and 12 weeks of absence from work. These represent non-complex cases of absence. Thus, these cases are not known to drive the most burden to the individual or the society. Is it possible to justify this criterion?

Authors’ response: We agree with the reviewer that the inclusion criterion (3 to 12 weeks of sickness absence) may result in non-complex cases of absence. However, the criterion was consciously chosen to create an opportunity to examine a group of young employees with CMD in the early stages before the cases become too complex. The idea was to explore the experiences of a group that has gained less attention in comparison to those with long-term sickness absence and make it possible to reflect upon early prevention or return to work. This has been clarified in line 212-215. 

Reviewer comment: In a qualitative study, participants’ characteristics and their context are usually described more in depth. Also, based on the Finnish study they mentioned, is it possible to add gender, and education? Also, what was the sick leave status of the participants? For managers, was the years of experience collected and how many employees under the supervision of the supervisors? This could influence the experience with CMD workers.

Authors’ response: In Table 1, the characteristics of the young participants are described in terms of age, gender, work sector, and living conditions during the pandemic. The sick leave status of the young employees was also described in Table 1, but clarify the heading is now changed from “Was during the pandemic” to “Sick leave status during the pandemic”. Regarding the young employee’s educational status this information was not collected systematically in the data collection and therefore cannot be accounted for.

For the managers, information has been added in Table 2 regarding years of experience. Concerning the number of employees under the managers’ supervision, the inclusion criterion for the managers was that they had to have experience from being the manager of at least one young employee on sick leave due to CMD. However, this did not exclude the possibility that the managers had experiences from managing more than one young employee with CMD in different organizations, thus making it difficult to account for the number of employees under their supervision.

Reviewer comment: Information on how participants were recruited should be added.

Authors’ response: Information regarding the recruitment process have been added in line 202-209 (for the recruitment of the young employees) and in line 241-247 (for the recruitment of the managers). 

Reviewer comment: In the data analysis section, the authors mentioned that “the utterances discussing different aspects of the pandemic were coded as Covid (…) who thereafter reanalysed the Covid data through conventional content analysis”. Based on this, if I understood correctly, only the content coded COVID was taken and analysed for this study. In the data collection section, it is mentioned that the interviews were done during the pandemic. Thus, it is unclear why COVID was coded if the experience is within a covid context. The risk is to assume that covid did not have an impact, just because someone failed to talk about covid at one point in the interview. We cannot assume this. More information on this is needed to assess possible bias during coding.

Authors’ response: We thank you for this remark. For the current manuscript, the full interview transcriptions were re-read by the first author to form an understanding of the context in its whole. The material coded as Covid was then compared to the full transcriptions, and consensus was found in the interpretation. Thereafter, the Covid data were reanalysed through conventional content analysis and created new codes of the data previously coded as Covid. This information has now been added in lines 268-273. 

To further clarify, the overall research project and the research question specific for this manuscript are now more thoroughly described in lines 150 to 158. The risk of someone failing to talk about covid was minimized by adding a specific question in the interview guide, which is now described in lines 158-160, and by consciously using probing questions to follow up the participants’ thoughts regarding the effects of the pandemic, which is now described in lines 160-162. The procedure of the analysis process has been developed in lines 266 – 274.

Reviewer comment: Is the coding emergent or based on a framework?

Authors’ response: The coding is emergent which has now been clarified in lines 262-265.

Reviewer comment: The authors mentioned that an interdisciplinary team of researchers did the analyses, please clarify which disciplines?

Authors’ response: The different disciplinary backgrounds of the research team have been clarified in line 858 to 859.

Reviewer comment: Another point needs clarification: How do managers know that the employee they supervise was absent for a CMD. Is the reason for absenteeism confidential in Sweden?

Authors’ response: In this study, an inclusion criterion for the managers was that they had to have had responsibility for young employees who they knew had been on sick leave due to CMD. In Sweden, employees are not obligated to account for the reason for their sick leave, but they can choose to share this information with their employer. This has been clarified in lines 251-252.

Reviewer comment: In the results section, it is not always possible to see what caused the absence, maintained the absence or made it difficult to return or sustain work. Young's (2010 in JSTOR) framework would help structure this information.

Authors’ response: We agree that the connections between the hindering or enabling factors and the young employees’ ability to either return to or remain at work were not always clear. We have therefore revised all parts of the results section to present these matters in a more clearly defined way. Thank you for this comment. 

Reviewer comment: Most of the results describe what most people had to live with during the pandemic. Thus, the mechanisms particular to young workers having CMD and supervisors having workers with CMD is not clear. Consequently, it is not clear what the manuscript adds to the existing knowledge.

Authors’ response: We agree that this was not always clear and have now revised all parts of the results section to clarify the mechanisms particular to young workers with CMD. These additions are visible in the tracked changes in the manuscript, for example in lines 465-468 and 504-507. 

Reviewer comment: The presentation of the results could be revised in order to present the results in more depth, among other things by looking more closely at the links between increased balance and decreased demands, increased demands at work and low staffing. Uncertainty appears to be in part underneath the work demands and low staffing. What these results appear to highlight is the concept of the margin of manoeuvers in work disability (see Durand et al. 2016 Handbook of return to work). It can thus be an enabling factor when a margin of manoeuvers exists or an obstacle when the margin of manoeuver is insufficient. Having a low margin of manoeuvers for managers may have contributed to the employee falling through cracks. The citation of IP 19 (lines 351-354) covers the themes of isolation, uncertainty and to some extent the boundaries. The 3 aspects are possibly linked, but the manuscript fails to provide a more in-depth understanding of how factors interrelate. This supports the need to add more links between factors. 

Authors’ response: 

Thank you for this useful reference, we have included the concept in the discussion section (lines 794-806). We have also revised the discussion section to reflect more clearly upon how the different hindering and enabling factors are linked and intertwined, and how they sometimes are two sides of the same coin. 

Reviewer comment: Concerning the distinction between work and family and the proximity of the family. Based on gender differences known in the literature, was the family proximity more problematic for men participants?

Authors’ response: No, we did not see any distinct gender differences

Reviewer comment: The citation lines 477-479 (“So, they have been strong anyway during the pandemic, and I think that is because we have tried to have a lot of dialogue, and I have tried to go out with weekly newsletters and clear information at … frequent, every week. Not too much and not too little, but what is absolutely necessary. And the staff knew that. (IP 14, female manager, head of unit, health care)) does appear to be a managerial strategy in general, and not in line with young workers having CMD. It is not always clear if managers talk about managing in a covid context or managing young workers having CMD in a covid context.

Authors’ response: Thank you for this comment, we agree that this was not always clear and have revised all parts of the results section to clarify that the managers talk about managing young employees with CMD in a covid context. These additions are visible in the tracked changes in the manuscript, for example in lines 525-527 and 558-561.

Reviewer comment: The discussion presents three interesting findings. However, these findings are not the main findings, as these did stand out in the results. As an example, in line 513 “….could imply difficulties for managers in understanding and recognising certain challenges for young people with CMD to remain at or return to work”, this was not clear for me after reading the results.

Authors’ response: Thank you for this important comment. We agree, and the discussion section as well as the conclusions have been re-written to connect more clearly to the results section. 

6. PLOS authors have the option to publish the peer review history of their article (what does this mean?). If published, this will include your full peer review and any attached files.

Do you want your identity to be public for this peer review? For information about this choice, including consent withdrawal, please see our Privacy Policy.

Reviewer #1: No

---

## [Decision Letter · Decision Letter 1]

15 Mar 2023

PONE-D-22-17806R1Hindering and enabling factors for young employees with Common Mental Disorder to remain at or return to work affected by the Covid-19 pandemic – a qualitative interview study with young employees and managersPLOS ONE

Dear Dr. Olsson,

Thank you for submitting your manuscript to PLOS ONE. After careful consideration, we feel that it has merit but does not fully meet PLOS ONE’s publication criteria as it currently stands. Therefore, we invite you to submit a revised version of the manuscript that addresses the points raised during the review process.

We look forward to receiving your revised manuscript.

Kind regards,

Nabeel Al-Yateem, PhD

Academic Editor

PLOS ONE

Journal Requirements:

Reviewers' comments:

Reviewer's Responses to Questions

**Comments to the Author**

1. If the authors have adequately addressed your comments raised in a previous round of review and you feel that this manuscript is now acceptable for publication, you may indicate that here to bypass the “Comments to the Author” section, enter your conflict of interest statement in the “Confidential to Editor” section, and submit your "Accept" recommendation.

Reviewer #1: All comments have been addressed

Reviewer #2: (No Response)

2. Is the manuscript technically sound, and do the data support the conclusions?

Reviewer #1: Yes

Reviewer #2: Yes

3. Has the statistical analysis been performed appropriately and rigorously? 

Reviewer #1: N/A

Reviewer #2: I Don't Know

4. Have the authors made all data underlying the findings in their manuscript fully available?

Reviewer #1: No

Reviewer #2: Yes

5. Is the manuscript presented in an intelligible fashion and written in standard English?

Reviewer #1: Yes

Reviewer #2: Yes

6. Review Comments to the Author

Reviewer #1: I'd like to commend the authors on their extensive revisions. This revised version was a pleasure to read. The results are easier to follow. The results are also very clear on the impact of COVID for young employee having CMD. In its current form, the manuscript does contribute to the body of knowledge.

I have only minor revisions to suggest:

Abstract : The conclusion should be in line with the results presented in the abstract. I suggest moving the findings in the conclusion in the results. The conclusion could be on the two-side of the same coin for hindering and enabling factors and interactions within the categories.

Results:

Line 297: there is an extra space to delete “…at home , and..”

Please add that in this study no gender differences were noted.

For the availability of the data, the authors provided a justification.

Reviewer #2: PLOS ONE

Hindering and enabling factors for young employees with Common Mental Disorder to remain at or return to work affected by the Covid-19 pandemic – a qualitative interview study with young employees and managers

Reviewers comments 13.3.23

Title is appropriate

Abstract p (page) 3

All clearly described and appropriate, if word limits allow, could the age range of the ‘young’ employees be included to help readers please, and are the managers and employees both young? Would be good just to clarify here.

1. Introduction p4

P4 line 72 Thus, societal changes e.g. were not as… could you add examples here

Line 76-slight repetition of earlier sentiments Even though Sweden did not instigate any lockdowns, the pandemic changed the ways work was carried out

P5 line 92 ‘when transferring to a digital context’ could you give a little explanation for this please

P5 line 108 ‘on the impact of age, gender, education, living alone and telework on occupational well-being found that well-being deteriorated more for the young participants compared with the other age groups [19]’. Could the authors say a little more directly why they ar

e focusing on young adults, and which ages groups are covered by the lit reviews please.

P5 112 ‘In addition, it was suggested that young people were potentially more likely to work in occupations where the working conditions were affected by the pandemic to a greater extent, such as human contact professions, including healthcare and service professions’.

This is a bit vague, would benefit from more specificity, and a reference

P6, line 119 ‘…2020, and data collection was conducted during the COVID-19 pandemic. Because the project collected 120 data regarding sick leave and return to work (RTW) – outcomes that most likely were affected by the 121 pandemic - the protocol was developed with a research question relating to the COVID-19 pandemic, 122 namely how the COVID-19 pandemic had affected the participants’ experiences. This research question 123 can provide an in-depth understanding of young employees’ and managers’ perspectives through a 124 qualitative approach, providing a comprehensive understanding of how the pandemic affected young 125 people with CMD and what lessons can be learned.’

Whilst this is well written, it could help to be more specific, experiences of ? perspectives of ? It’s a bit vague at the moment

2. Materials and methods p6

Line 133 ‘The full scope of the larger project is described in a published study protocol [20].’

It would be good to have a clear, short, outline of the study, and to understand what the specific research question underpinning this sub-project was.

Line 135, what does the Tong paper describe?

Data collection

P6, more detail is needed here about how employees and managers were recruited, from which industries, which age brackets, how they were approached.

P6- ‘A semistructured interview guide was created based on the aim of the larger ongoing study’

how were the themes, questions decided, can you give some overview of the topics included please? It’s an interesting study, I’d like to know more about how it was constructed, and the specifics of what it was exploring, and how it was analysed.

P7 ‘…affected their ability to remain at work’ could the team clarify here please, e.g. between working from home or working from the office?

Data analysis

Could the team provide more specific information about how the analysis was conducted please, e.g. how were the codes decided on e.g. Grounded theory; Interpretative phenomenological analysis; and Framework analysis,

Open, closed, or a mix of codes relating to areas of study, conditions?

What was the results, how was it refined?

Where there different codes used, or foci for each cohort -managers, employees? Did the team use any theoretically informed frameworks?

3. Results p10

It would be helpful to have a sense of context, and to know what types of jobs participants were engaging in, to understand how this may have influenced perceptions about mental health.

The paper is very well written, its just difficult at the moment to get a full sense of the specifics of how the analysis was focused, completed, and how the findings arose, its very polished, personally-I really like to get a sense of the messiness of the findings, where the tensions, gaps, inconsistencies occur.

Where there any similarities/differences between accounts based on age, types of industry, locality, gender experience? These may all not be relevant, but it would be good to see more data, and context and specificity coming through. Where were the boundaries, was anything grey, or liminal? What was contested, or ignored?

The quotes are great, I like to see more show, and less tell- slightly less polished, more depth on the complex areas-negotiations. What does poorer mental health look like in a pandemic when others are suffering too, what if your job is classified as a key role, and you carry on, but feel greater belonging, but worse in other respects?

4. Discussion p21

Well written, I’d like to see the findings engage with wider societal issues, what might some implications be for government employment policies, mental health and employment legislation, support? Is there a perceived need for focusing on bespoke health policies for younger employees- in a climate of rising costs, who funds this, and what is the role /responsibilities of managers?

Thank-you for letting me review this paper.

Overall a very well written paper, clear, easy to follow, excellent English.

I’d like to see more direct and specific information about how the study was constructed, how people were selected, and about the analysis and findings decision processes, I’d like to see more of the tensions, troubled boundaries, more data, the ‘grey’ areas laid bare for discussion and wider implications.

7. PLOS authors have the option to publish the peer review history of their article (what does this mean?). If published, this will include your full peer review and any attached files.

Reviewer #1: **Yes: **Marie-France Coutu

Reviewer #2: No

---

## [Author Response · Author response to Decision Letter 1]

5 Apr 2023

PONE-D-22-17806R1

Hindering and enabling factors for young employees with Common Mental Disorder to remain at or return to work affected by the Covid-19 pandemic – a qualitative interview study with young employees and managers

PLOS ONE

Journal Requirements:

Author’s comment: We have double checked and the reference list is complete and correct. 

Reviewers' comments:

Reviewer's Responses to Questions

Comments to the Author

1. If the authors have adequately addressed your comments raised in a previous round of review and you feel that this manuscript is now acceptable for publication, you may indicate that here to bypass the “Comments to the Author” section, enter your conflict of interest statement in the “Confidential to Editor” section, and submit your "Accept" recommendation.

Reviewer #1: All comments have been addressed

Reviewer #2: (No Response)

2. Is the manuscript technically sound, and do the data support the conclusions?

Reviewer #1: Yes

Reviewer #2: Yes

3. Has the statistical analysis been performed appropriately and rigorously? 

Reviewer #1: N/A

Reviewer #2: I Don't Know

4. Have the authors made all data underlying the findings in their manuscript fully available?

Reviewer #1: No

Reviewer #2: Yes

5. Is the manuscript presented in an intelligible fashion and written in standard English?

Reviewer #1: Yes

Reviewer #2: Yes

6. Review Comments to the Author

Note: When the authors response refers to specific lines in the Tracked changes manuscript, it refers to the line numbers when the changes are made visible

Reviewer comment: Reviewer #1: I'd like to commend the authors on their extensive revisions. This revised version was a pleasure to read. The results are easier to follow. The results are also very clear on the impact of COVID for young employee having CMD. In its current form, the manuscript does contribute to the body of knowledge.

Authors’ response: We are glad to hear that you think the manuscript has improved and we would like to thank you for your review that contributed to the improvements!

I have only minor revisions to suggest:

Reviewer comment: Abstract : The conclusion should be in line with the results presented in the abstract. I suggest moving the findings in the conclusion in the results. The conclusion could be on the two-side of the same coin for hindering and enabling factors and interactions within the categories.

Authors’ response: we have rewritten the abstract in accordance with your suggestion. 

Results:

Reviewer comment: Line 297: there is an extra space to delete “…at home , and..” 

Authors’ response: Thank you, we deleted the extra space.

Reviewer comment: Please add that in this study no gender differences were noted. 

Authors’ response: 

The sentence: “Gender differences were not reported” was added, on line 277-278 

Reviewer comment: For the availability of the data, the authors provided a justification.

Authors’ response: Thank you

Reviewer #2: PLOS ONE

Hindering and enabling factors for young employees with Common Mental Disorder to remain at or return to work affected by the Covid-19 pandemic – a qualitative interview study with young employees and managers

Reviewers comments 13.3.23

Title is appropriate

Abstract p (page) 3

Reviewer comment: All clearly described and appropriate, if word limits allow, could the age range of the ‘young’ employees be included to help readers please, and are the managers and employees both young? Would be good just to clarify here.

Authors’ response: Thank you, we have changed the order to make it easier to read and added the age span of the young employees.

Reviewer comment: 1. Introduction p4 P4 line 72 Thus, societal changes e.g. were not as… could you add examples here

Authors’ response: The line referred to the examples above, but as it was not necessary.

Reviewer comment: Line 76-slight repetition of earlier sentiments Even though Sweden did not instigate any lockdowns, the pandemic changed the ways work was carried out.

Authors’ response: Thank you for this comment; to improve readability the sentence has been removed.

Reviewer comment: P5 line 92 ‘when transferring to a digital context’ could you give a little explanation for this please

Authors’ response: Thank you for this comment, we added a sentence to clarify (line 98-99). 

Reviewer comment: P5 line 108 ‘on the impact of age, gender, education, living alone and telework on occupational well-being found that well-being deteriorated more for the young participants compared with the other age groups [19]’. Could the authors say a little more directly why they are focusing on young adults, and which ages groups are covered by the lit reviews please.

Authors’ response: We have added information on the participants age range in the referred studies in the lit review.

Reviewer comment: P5 112 ‘In addition, it was suggested that young people were potentially more likely to work in occupations where the working conditions were affected by the pandemic to a greater extent, such as human contact professions, including healthcare and service professions’. This is a bit vague, would benefit from more specificity, and a reference

Authors’ response: To clarify that the entire content of the paragraph refers to the same study, an addition has been made in line 117

Reviewer comment: P6, line 119 ‘…2020, and data collection was conducted during the COVID-19 pandemic. Because the project collected 120 data regarding sick leave and return to work (RTW) – outcomes that most likely were affected by the 121 pandemic - the protocol was developed with a research question relating to the COVID-19 pandemic, 122 namely how the COVID-19 pandemic had affected the participants’ experiences. This research question 123 can provide an in-depth understanding of young employees’ and managers’ perspectives through a 124 qualitative approach, providing a comprehensive understanding of how the pandemic affected young 125 people with CMD and what lessons can be learned.’

Whilst this is well written, it could help to be more specific, experiences of ? perspectives of ? It’s a bit vague at the moment

Authors’ response: We thank you for this remark. We have re-written the section on the relation between the original project and the current study (line 129-133 and 149-157).

Reviewer comment: 2. Materials and methods p6

Line 133 ‘The full scope of the larger project is described in a published study protocol [20].’

It would be good to have a clear, short, outline of the study, and to understand what the specific research question underpinning this sub-project was.

Authors’ response: Thank you for this important comment. We have added a paragraph to clarify (Line 149-157)

Reviewer comment: Line 135, what does the Tong paper describe?

Authors’ response: The Coreq checklist (by Tong) add quality to qualitative studies by setting standard in three different domains, namely: Research team and reflexivity, Study design, analysis and findings. By completing the checklist, the researchers demonstrate how they have reported on different items within the domains, for example what sample methods were used or how many participants were included.

Reviewer comment: Data collection

P6, more detail is needed here about how employees and managers were recruited, from which industries, which age brackets, how they were approached.

Authors’ response: We have added information about the recruitment process, line 195-196 line 202-203, 204-207 and 192-195.

Reviewer comment: P6- ‘A semistructured interview guide was created based on the aim of the larger ongoing study’ how were the themes, questions decided, can you give some overview of the topics included please? It’s an interesting study, I’d like to know more about how it was constructed, and the specifics of what it was exploring, and how it was analysed.

Authors’ response: We added more information about the interview guides on the lines 166-167 and 169-170. We also think that the changes made under Materials and methods in line 149-157 helps to clarify. 

Reviewer comment: P7 ‘…affected their ability to remain at work’ could the team clarify here please, e.g. between working from home or working from the office?

Authors’ response: A clarification has been made in line 176-177

Reviewer comment: Data analysis

Could the team provide more specific information about how the analysis was conducted please, e.g. how were the codes decided on e.g. Grounded theory; Interpretative phenomenological analysis; and Framework analysis,

Open, closed, or a mix of codes relating to areas of study, conditions

What was the results, how was it refined?

Where there different codes used, or foci for each cohort -managers, employees? Did the team use any theoretically informed frameworks?

Authors’ response: we have added more information on the initial coding and analysis process on lines 260-267, 270

Reviewer comment: 3. Results p10

It would be helpful to have a sense of context, and to know what types of jobs participants were engaging in, to understand how this may have influenced perceptions about mental health.

The paper is very well written, its just difficult at the moment to get a full sense of the specifics of how the analysis was focused, completed, and how the findings arose, its very polished, personally-I really like to get a sense of the messiness of the findings, where the tensions, gaps, inconsistencies occur.

Where there any similarities/differences between accounts based on age, types of industry, locality, gender experience? These may all not be relevant, but it would be good to see more data, and context and specificity coming through. Where were the boundaries, was anything grey, or liminal? What was contested, or ignored?

The quotes are great, I like to see more show, and less tell- slightly less polished, more depth on the complex areas-negotiations. What does poorer mental health look like in a pandemic when others are suffering too, what if your job is classified as a key role, and you carry on, but feel greater belonging, but worse in other respects?

Authors’ response: The study has included participants from many different professions and industries to obtain as rich a material as possible. This means that the different experiences are not shared by everyone but shed light on the phenomenon from the individual participant's perspective and experience. Conventional content analysis is a suitable method for sorting these experiences into categories and subcategories so that patterns in the data emerge. It may look tidy but in practice it means that different voices are heard. Other analysis method would perhaps have been more suitable for showing conflicts in the material and we will consider other methods in the future. 

Reviewer comment: 4. Discussion p21

Well written, I’d like to see the findings engage with wider societal issues, what might some implications be for government employment policies, mental health and employment legislation, support? Is there a perceived need for focusing on bespoke health policies for younger employees- in a climate of rising costs, who funds this, and what is the role /responsibilities of managers?

Authors’ response: Thank you for this remark. We have added a paragraph on the employer’s role and responsibility in the introduction section (line 122-125)

Thank-you for letting me review this paper.

Overall a very well written paper, clear, easy to follow, excellent English.

I’d like to see more direct and specific information about how the study was constructed, how people were selected, and about the analysis and findings decision processes, I’d like to see more of the tensions, troubled boundaries, more data, the ‘grey’ areas laid bare for discussion and wider implications. 

Authors’ response: Thank you for reviewing our paper and for all useful comments that will improve our manuscript.

7. PLOS authors have the option to publish the peer review history of their article (what does this mean?). If published, this will include your full peer review and any attached files.

Do you want your identity to be public for this peer review? For information about this choice, including consent withdrawal, please see our Privacy Policy.

Reviewer #1: Yes: Marie-France Coutu

Reviewer #2: No

---

## [Decision Letter · Decision Letter 2]

24 May 2023

Hindering and enabling factors for young employees with Common Mental Disorder to remain at or return to work affected by the Covid-19 pandemic – a qualitative interview study with young employees and managers

PONE-D-22-17806R2

Dear Dr. Olsson,

We’re pleased to inform you that your manuscript has been judged scientifically suitable for publication and will be formally accepted for publication once it meets all outstanding technical requirements.

Kind regards,

Nabeel Al-Yateem, PhD

Academic Editor

PLOS ONE

Additional Editor Comments (optional):

Reviewers' comments:

Reviewer's Responses to Questions

**Comments to the Author**

1. If the authors have adequately addressed your comments raised in a previous round of review and you feel that this manuscript is now acceptable for publication, you may indicate that here to bypass the “Comments to the Author” section, enter your conflict of interest statement in the “Confidential to Editor” section, and submit your "Accept" recommendation.

Reviewer #1: All comments have been addressed

Reviewer #2: (No Response)

2. Is the manuscript technically sound, and do the data support the conclusions?

Reviewer #1: Yes

Reviewer #2: Yes

3. Has the statistical analysis been performed appropriately and rigorously? 

Reviewer #1: N/A

Reviewer #2: Yes

4. Have the authors made all data underlying the findings in their manuscript fully available?

Reviewer #1: No

Reviewer #2: No

5. Is the manuscript presented in an intelligible fashion and written in standard English?

Reviewer #1: Yes

Reviewer #2: Yes

6. Review Comments to the Author

Reviewer #1: The authors have responded to all my comments.

For Q4 "Have the authors made all data underlying the findings in their manuscript fully available?" The answer is no, as the data cannot be made public due to ethical constraints.

Reviewer #2: Hindering and enabling factors for young employees with Common Mental Disorder to

remain at or return to work affected by the Covid-19 pandemic – a qualitative interview

study with young employees and managers

appropriate title

Abstract

Fine

1. Intro

Improved, its easier to understand the connections and the aim.

2. Materials and methods

Improved, clearer

3. Results

Much clearer, more direct, easier to navigate. Personally, I feel this section could be reduced in length, as there is some repetition, and I’d like to see a few more quotes to illustrate the data.

4. Discussion

This is better, at the moment there is too much focus on findings, it would benefit from being shorter, with more consideration about the wider implications, e.g. for employees, for sickness management, for employers, the economy? It doesn’t need to be re-written, just edited.

It sounds like more flexible working practices are called for when managing short term CMD leave? And perhaps that work, home boundaries become blurred, but that this has benefits and consequences, perhaps for sickness leave policies, insurance, practices, perceptions about mental illness?

5. Conclusion- a slightly refocused, wider implications discussion section should provide a couple of bigger points to consider

Once 4 and 5 have been revised, I'm happy to recommend for publication.

7. PLOS authors have the option to publish the peer review history of their article (what does this mean?). If published, this will include your full peer review and any attached files.

Reviewer #1: **Yes: **Marie-France Coutu

Reviewer #2: No

---

## [Editor Report · Acceptance letter]

29 May 2023

PONE-D-22-17806R2 

Hindering and enabling factors for young employees with Common Mental Disorder to remain at or return to work affected by the Covid-19 pandemic – a qualitative interview study with young employees and managers 

Dear Dr. Olsson:

I'm pleased to inform you that your manuscript has been deemed suitable for publication in PLOS ONE. Congratulations! Your manuscript is now with our production department. 

Kind regards, 

on behalf of

Dr. Nabeel Al-Yateem 

Academic Editor

PLOS ONE